# A protocol and novel tool for systematically reviewing the effects of mindful walking on mental and cardiovascular health

**Dustin W. Davis** *, **Bryson Carrier**, **Brenna Barrios**, **Kyle Cruz**, **James W. Navalta**

Department of Kinesiology and Nutrition Sciences, University of Nevada, Las Vegas, Las Vegas, Nevada, United States of America

☯ These authors contributed equally to this work.
* Dustin.davis@unlv.edu

**Funding:** The author(s) received no specific funding for this work.

## Abstract

To our knowledge, no published systematic review has described the effects of mindful walking on mental and cardiovascular health. We have aimed to fill this gap by first establishing our systematic review protocol. Our protocol was adapted from the Preferred Reporting Items for Systematic Reviews and Meta-Analyses (PRISMA) and is registered in PROSPERO (Registration Number: CRD42021241180). The protocol is described step-by-step in this paper, which we wrote to achieve three objectives: to adhere to the best practices stated in the PRISMA guidelines, to ensure procedural transparency, and to enable readers to co-opt our protocol for future systematic reviews on mindful walking and related topics. To achieve our third objective, we provide and explain a novel tool we created to track the sources we will find and screen for our review. Ultimately, the protocol and novel tool will lead to the first published systematic review about mindful walking and will also facilitate future systematic reviews.

## Introduction

Much of the global population suffers from poor mental and cardiovascular health. The prevalence of mental disorders among adults is 17.6% and 29.2% over one-year and a lifetime, respectively [1]. The leading cause of death among adults is cardiovascular diseases (CVD) [2]. Adults' mental and cardiovascular health may be worsening because of the obesity epidemic. Globally, 10.8% and 14.9% of adult males and females have obesity, respectively [3]. The epidemic is worse in high-income countries such as the United States, where 42.4% of adults have obesity and another 9.2% have severe obesity [4]. Obesity positively correlates with anxiety [5] and depression [6] and increases the risk of CVD and CVD-related deaths [7]. The prevalence of mental disorders, CVD, and obesity has grown despite new medical knowledge and medications. This paradox has prompted researchers to study unconventional treatments. One of these treatments is mindful exercise, defined as "physical exercise executed with a profound inwardly directed contemplative focus" [8–10]. Whether mindful exercise eases the burden of poor mental and cardiovascular health is not yet known.

**Competing interests:** The authors have declared that no competing interests exist.

Published literature about mindful exercise is focused on limited populations and interventions. Two meta-analyses concluded that mindful exercise reduces short-term depression [11] in people with clinical depression and increases sensorimotor function in patients after a stroke [12]. A third meta-analysis concluded that yoga, a specific type of mindful exercise, reduces anxiety more than non-mindful exercise [13]. But research has not yet explained if mindful exercises benefit broad populations. One mindful exercise that may be widely accessible is mindful walking. Walking does not require the physical skills or training of other common mindful exercises such as tai chi, qigong, or yoga. Walking also improves mental [14] and cardiovascular health [15] independent of mindful practices. It is unclear if mindful walking improves these aspects of health better than non-mindful walking. A prerequisite to answering this question is characterizing the published literature about mindful walking.

To our knowledge, no published paper has reviewed the effects of mindful walking on mental and cardiovascular health. We aim to fill this gap by systematically reviewing the relevant literature. We selected the "systematic review" as our methodology after reading published guidance on review types [16, 17]. The primary aim of our systematic review will be to find and describe the mindful walking protocols, populations, and outcomes reported in published and gray literature. According to the Preferred Reporting Items for Systematic Reviews and Meta-Analyses (PRISMA) guidelines, the best practices for performing a systematic review include publishing the protocol independent of the review to ensure procedural transparency [18]. This paper and the PRISMA-P Checklist (S1 File) show that our work aligns with the best practices. Alignment with the best practices has been confirmed by the first author, who will also be the guarantor of the systematic review. As our systematic review of mindful walking will be, to our knowledge, the first published on the topic, it is especially important to publish our protocol. That protocol is described in this paper to achieve three objectives: 1) To adhere to the best practices stated in the PRISMA guidelines; 2) To ensure procedural transparency; and 3) To enable readers to co-opt our protocol for future systematic reviews on mindful walking and related topics. To achieve our third objective, we explain a novel tool we created to track the sources we will find and include in our review.

## Materials and methods

Developing the systematic review protocol began with the first author performing two scoping searches in Google Scholar and PubMed on two separate days within one week. The scoping searches suggested that there were no published systematic reviews of mindful walking. Then, the first author consulted peer-reviewed guidance [16, 17] and a book [19] about conducting systematic reviews in health fields. The findings of the scoping search and guidance were presented to the full review team. The team deliberated and agreed upon the protocol presented in this article. The details of the protocol are also available in PROSPERO (Registration Number: CRD42021241180). The protocol has seven sequential steps that create the flow of our systematic review: 1) Create a clear review question and Participant, Intervention, Comparator, Outcome, Study Design (PICOS) criteria; 2) Create eligibility criteria (inclusion and exclusion criteria); 3) Create and follow a search strategy; 4) Document sources that are included and excluded according to the eligibility criteria; 5) Assess the sources included in the systematic review for risk of bias; 6) Extract the pertinent data from the included full-text articles and write a narrative synthesis; and 7) Disseminate the findings of the systematic review. An optional part of systematic reviews that we do not plan to conduct is a meta-analysis. We do not plan to conduct a meta-analysis because our scoping searches showed that the relevant studies provided different mindful walking interventions, sampled different populations, used different comparison groups, and measured different outcomes in different settings. These

differences indicate clinical heterogeneity among the studies we expect to include, and clinical homogeneity is required for a valid meta-analysis [19]. The following subsections describe the seven steps of the protocol.

## Step 1: Create a clear review question and PICOS criteria

Step 1 is to create a clear review question and PICOS criteria (Table 1). The review question includes the phrase "meditative and mindful walking" because mindfulness is a key feature of the deliberate practice of meditative walking. Omitting "meditative walking" from the review question and search terms would risk missing studies of meditative walking, which innately involves walking and mindfulness. Because literature about mindful walking is scarce, we set broad PICOS criteria to capture as many relevant studies as possible. Studies with adults, regardless of mental or cardiovascular disease status, will be relevant as long as the adults completed any form of meditative or mindful walking. Controlled, uncontrolled, randomized, nonrandomized, parallel, and crossover studies will be considered. Information about participants' characteristics, mental and cardiovascular health, and the setting in which the participants walked will be collected.

## Step 2: Create eligibility criteria

Step 2 is to create inclusion and exclusion criteria, collectively called eligibility criteria. The review question and PICOS criteria should form the basis of a systematic review's eligibility criteria, which should be explicit and succinct. These qualities give clarity to the review team members, enabling them to exclude irrelevant sources during the screening process. An expeditious screening process is important to the systematic review because review teams may be

**Table 1. Review question and PICOS table.**

| | |
|---|---|
| **Review Question** | What is the evidence for meditative and mindful walking as therapies for improving mental and cardiovascular health in adults with and without psychological disorders or cardiovascular diseases? |
| **Population** | Adults with or with no psychological disorders or cardiovascular diseases<br>• Will extract participants' age, sex, gender, nationality, disease status, medication use, and history of meditation or mindfulness practice |
| **Intervention** | Meditative walking or mindful walking<br>• Any form of walking with a meditative or mindful component used to reduce anxiety or depression, increase mindfulness, or improve cardiovascular risk factors<br>• Operational definition of meditative and mindful walking: Walking with an inwardly directed mental focus and a concentration on muscular movements, body alignment, and/or breath<br>• Will extract the frequency, intensity, type, duration, and location (e.g., indoors, outdoors) of the intervention |
| **Comparator** | Placebo or negative control in controlled studies<br>No comparator in uncontrolled studies |
| **Outcomes** | Any beneficial or adverse changes in any quantitative measure of anxiety, depression, mindfulness, or cardiovascular health or risk<br>• Any subjective self-reported measures of anxiety, depression, or mindfulness<br>• Any objective cardiovascular biomarkers |
| **Setting** | Any physical environment (indoors, outdoors, urban, rural, built-up, or natural) |
| **Study Design** | Only studies with interventions, and no observational studies<br>• Controlled or uncontrolled<br>• Randomized or nonrandomized<br>• Crossover design (participants complete the intervention and control arms) or parallel design (participants complete only the intervention or control arm) |

**Table 2. Eligibility criteria.**

| | |
|---|---|
| **Participants** | Adults of any age, sex, gender, nationality, disease status, medication use, and history of meditation or mindfulness practice |
| **Inclusion Criteria** | 1. The source is a published article in a peer-reviewed journal or is an unpublished or published master's thesis or doctoral dissertation<br>2. The source is written in English<br>3. The source reports the findings of an interventional study<br> a. The intervention is any walking with a meditative or mindful component used to reduce anxiety or depression, increase mindfulness, or improve cardiovascular risk factors<br> b. At least one reported outcome is a measure of anxiety, depression, mindfulness, or cardiovascular health |
| **Exclusion Criteria** | 1. The source is not a published, peer-reviewed journal article or a published or unpublished master's thesis or doctoral dissertation<br>2. The source is written in any language other than English<br>3. The source reports the findings of an interventional study with an intervention or outcomes irrelevant to this systematic review<br> a. There is a walking intervention without a meditative or mindful component<br> b. None of the reported outcomes are a measure of anxiety, depression, mindfulness, or cardiovascular health<br>4. The source reports the findings of an observational study (i.e. there is no walking intervention) |

required to screen hundreds to thousands of sources. We consulted our review question and PICOS criteria to create our eligibility criteria (Table 2). The broad PICOS criteria led to broad eligibility criteria. Our inclusion criteria will allow the inclusion of unpublished master's theses and doctoral dissertations. This decision was made to capture as much data about mindful walking interventions as possible. Because we are interested in interventions, our exclusion criteria will demand the exclusion of observational studies.

## Step 3: Create and follow a search strategy

After confirming the eligibility criteria, the review team should follow Step 3 by creating and following a search strategy (Table 3). Consistent with our broad PICOS and eligibility criteria, we will search in five databases for all relevant studies, regardless of publication year. The only string searched in each database will be the search combination (Table 3). The search combination was created by the first author after conducting the scoping searches described in the Materials and methods. Then, the full review team met by videoconference to test the search combination in each database. An iterative process of revision ensued until the search combination returned a manageable number of hits in each database (about 200–1,000 hits).

With the operational search combination established, the first author assigned roles to the five review team members. Two members formed Team A, two members formed Team B, and a fifth member assumed a solo role to settle disputes within teams over eligibility (Fig 1). The two members of Team A will search in Academic Search Premier, APA PsycInfo, PubMed, and SPORTDiscus. The two members of Team B will search in Google Scholar. This assignment of databases will divide the workload of the search nearly equally between the two teams. Piloting the search combination (Table 3) showed that Team A's four databases returned approximately 1,000 hits in total. Team B's one database, Google Scholar, returned more hits but only shows 100 pages that each contain 10 hits ($100 \times 10 = 1{,}000$ visible hits) [20]. Therefore, both teams will screen approximately 1,000 sources by title (the first step of the screening process). We named the search, screening process, and inclusion process the "search flow."

**Table 3. Search strategy.**

| Investigators | Team A: DD and BC | |
| --- | --- | --- |
| | Team B: BB and KC | |
| | Arbiter: JN | |
| Techniques | Search research databases for sources, including them in four stages:<br>1. Include sources by title<br>2. Include sources by abstract<br>3. Include sources by full text<br>4. Include sources from the reference lists of sources included by full text (journal articles, master's theses, and doctoral dissertations) | |
| Databases | Academic Search Premier, APA PsycInfo, Google Scholar, PubMed, and SPORTDiscus | |
| Included Types of Literature | Published, peer-reviewed journal articles; unpublished and published master's theses and doctoral dissertations | |
| Publication Date Range | No limit | |
| **Intervention Search Terms** | **Outcome Search Terms** | |
| "Meditative walk*" | "Stress" | "Cardiovascular" |
| "Walk* meditat*" | "Anxiety" | "Hypertens*" |
| "Mindful* walk*" | "Depress*" | "Blood pressure" |
| "Buddhis* walk*" | "Mindfulness" | "Cholesterol" |
| | "Health" | "Hyperglycem*" |
| | "Fitness" | "Blood sugar" |
| | "Allostatic load" | "Insulin*" |
| | "Disease" | |
| Search Combination | ((Meditative walk*) OR (walk* meditat*) OR (mindful* walk*) OR (Buddhis* walk*)) AND (stress OR anxiety OR depress* OR mindfulness OR health OR fitness OR allostatic load OR disease OR cardiovascular OR hypertens* OR blood pressure OR cholesterol OR hyperglycem* OR blood sugar OR insulin*) | |

Our search flow will funnel an initially extensive collection of sources into progressively smaller collections (Fig 2). The search flow is explained in Step 4.

### Step 4: Document sources that are included and excluded. Consider using our novel tool

Step 4 is the practical application of the systematic review protocol, the search flow (Fig 3). Our search flow is a process modeled on the 2009 PRISMA statement [18]. The search flow contains four distinct steps: 1) Identify relevant sources by their titles, 2) Screen sources by their abstracts, 3) Assess and include sources by their full texts, and 4) Include eligible sources found in the references of full texts included in Step 3. During the search flow, it will be imperative that we document sources' inclusion and exclusion clearly [17, 19]. Clear documentation

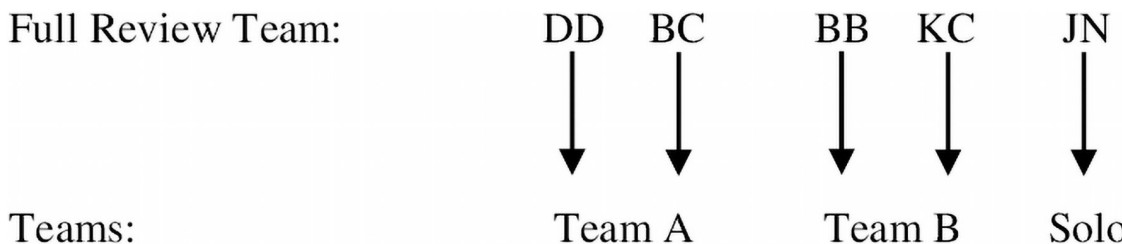

**Fig 1. The full review team was divided into smaller teams to search the databases.**

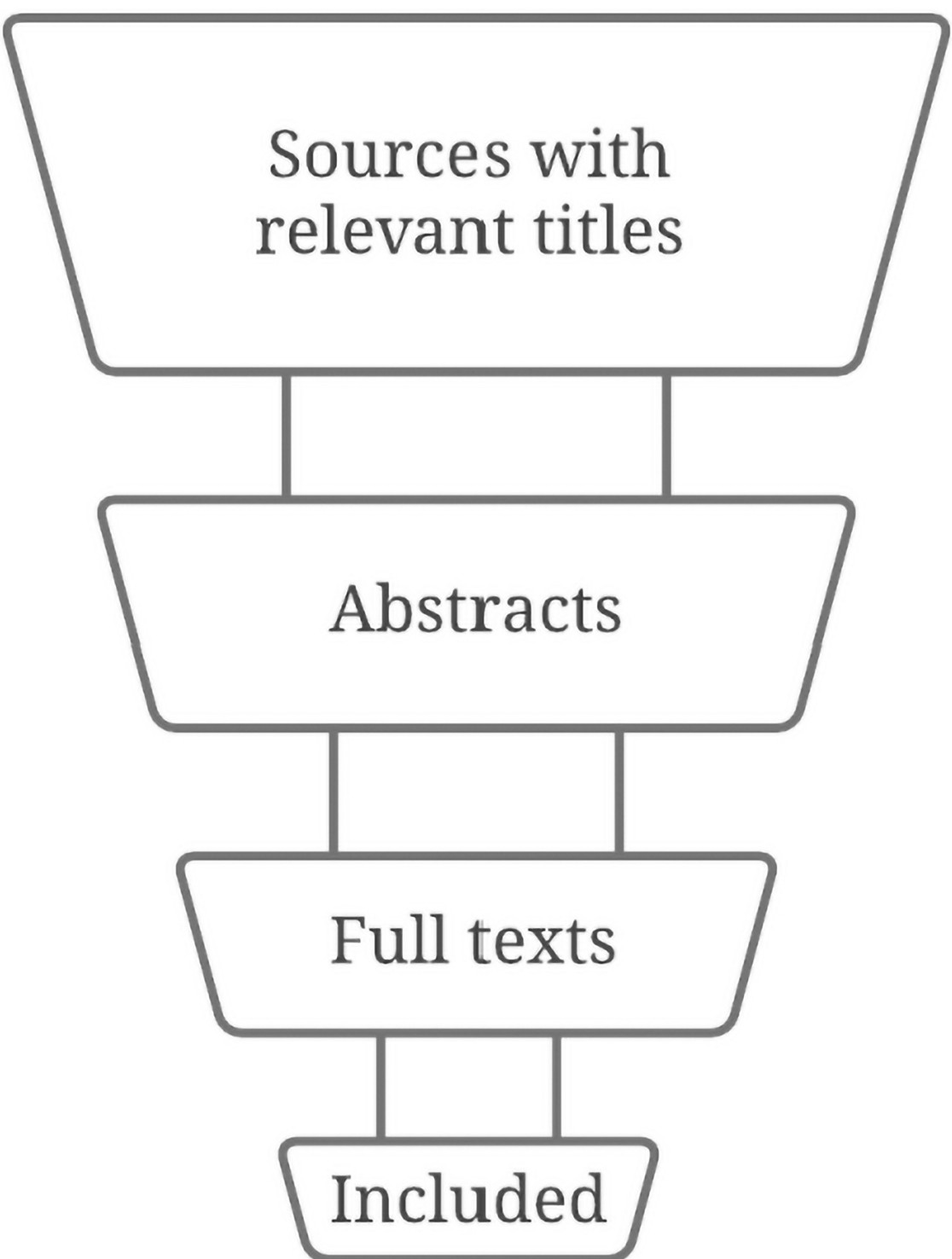

**Fig 2. The search flow will funnel sources into progressively smaller collections.**

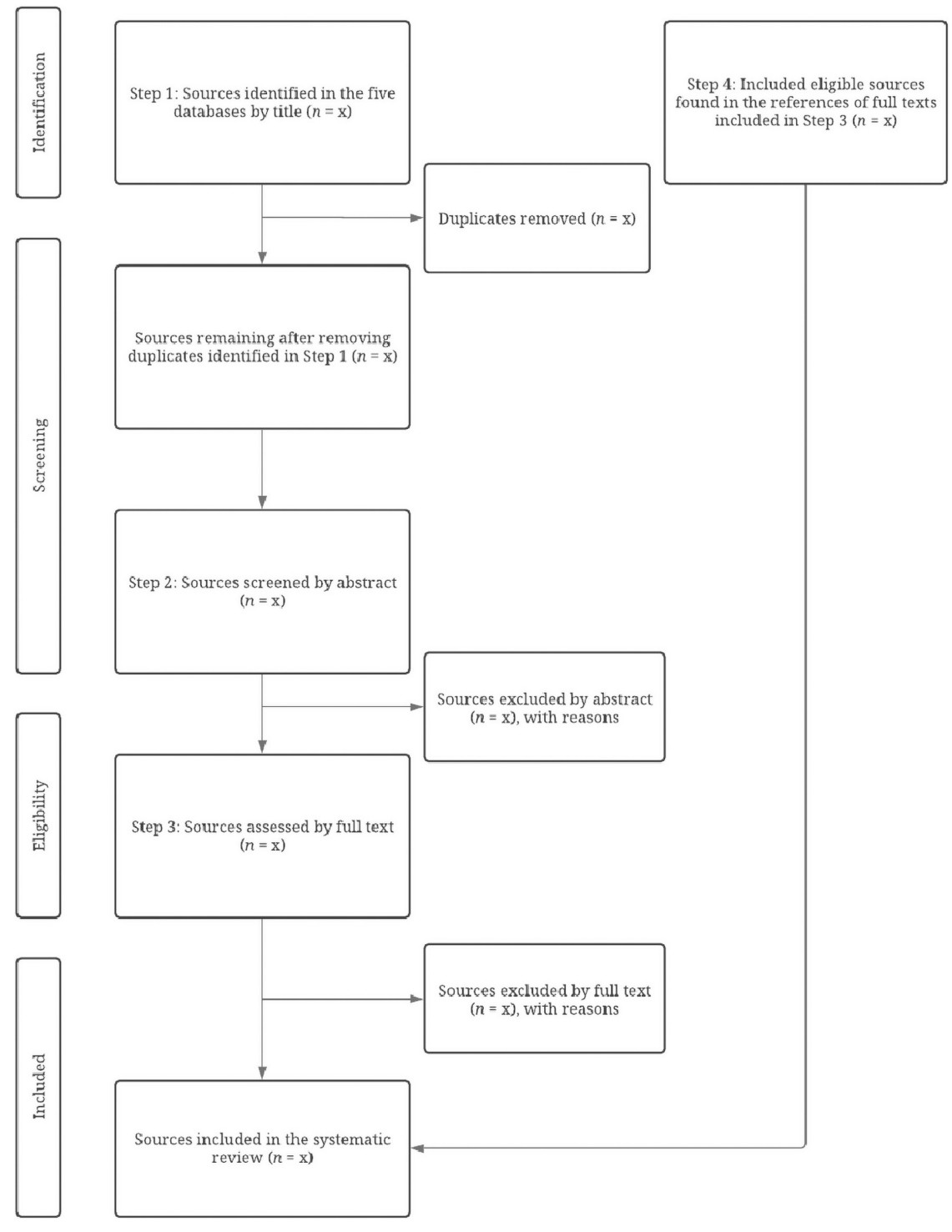

**Fig 3. Search flow.**

makes our systematic review transparent and reproducible. Reproducibility is a hallmark of systematic reviews that distinguishes them from traditional literature reviews (15).

To make our search flow reproducible, we created a novel tool in Google Sheets. The tool is a spreadsheet for Teams A and B to coordinate with each other and the solo member. The spreadsheet has five separate sheets, one for each member of Teams A and B and one for the solo member. During Step 1 of the search flow, members of Team A and B will enter three types of values into the sheet: 1) the number of hits each database returns, 2) the number of sources deemed relevant by title, and 3) the number of duplicate sources identified across the databases (identical sources found in ≥ 1 database). The sheet will automatically sum these values. The sheet is thus a precise record of members' progression through Step 1 of the search flow. The sheet also helps members record Steps 2–4. We explain all four steps of the search flow in Comprehensive Instructions for Completing the Search Flow (S2 File). The novel tool we created and used to track sources through the search flow is provided as a supplement called Novel Tool for Tracking the Inclusion and Exclusion of Sources (S3 File).

## Step 5: Assess the sources included in the systematic review for risk of bias

Step 5 is important for all systematic reviews and involves assessing the included sources' risk of bias [17, 19]. Knowing the risk of bias allows the reader to draw conclusions that align with the quality and strength of the evidence for an intervention affecting an outcome [17]. In our systematic review, we will assess sources' risk of bias (bias at the study-level) using tools specific to their respective study designs. Randomized, parallel trials will be assessed by using the revised Cochrane risk of bias tool for randomized trials (RoB 2) [21, 22]. Randomized, crossover trials will be assessed by using the RoB 2 or crossover trials [23, 24]. And non-randomized trials will be assessed by using the risk of bias in non-randomized studies—of interventions (ROBINS-I) tool [25, 26]. The risk of bias scores from these tools will be discussed in the narrative synthesis described in the next subsection.

## Step 6: Extract the pertinent data from the included full-text articles and write a narrative synthesis

Step 6 is to extract the pertinent data from the included full-text articles and write a narrative synthesis. For our review, Team A will extract the data from the first half of the included full texts, and Team B will extract the data from the second half of the included full texts. Within each team, the team members will verify that the data each member extracted are consistent. Disagreements will be discussed and resolved by reaching a consensus. The data that will be collected for our systematic review are the study designs, participants' characteristics, outcomes related to mental and cardiovascular health, and the setting in which the mindful walking interventions occurred. These data and information about the articles' authors, publication year, and country of origin will be entered into a spreadsheet. Once all data are in the spreadsheet, the first author will write a narrative synthesis of the data and report the main findings and implications of the systematic review. The narrative synthesis will also explain the authors' confidence in the cumulative evidence for mindful walking. That confidence will be based on the statistical significance and magnitude of the findings and the sources' risk of bias scores.

## Step 7: Disseminate the findings of the systematic review

Step 7 is the final step. After the data extraction and narrative synthesis of the systematic review are completed, the main findings and implications should be disseminated. We intend to complete the systematic review by July 2021 and submit the narrative synthesis for publication in a peer-reviewed academic journal by September 2021.

## Results and discussion

To our knowledge, our systematic review will be the first to describe the effects of mindful walking on mental and cardiovascular health. Therefore, our review will fill an important gap in the literature and form the cornerstone of future research on mindful walking. We are presenting our protocol to earn readers' trust in our review and to give them a framework to conduct their own systematic reviews. Our novel tool (the spreadsheet) for tracking the screening of articles will help researchers stay organized during their systematic reviews. While full-featured computer programs are available for conducting systematic reviews, many of the programs require a paid subscription. The best feature of our tool is that it is free. Any researcher who can access a spreadsheet program can use our tool. For this reason, our tool is also modifiable. Researchers can edit the columns, rows, color schemes, and cell formulas to meet their team's individual needs. This flexibility will surely be an asset to research teams that are unfamiliar with systematic review programs or are financially limited. Financial limitations are a major barrier to research teams in developing countries. In possession of our tool, researchers in these countries may be better able to conduct systematic reviews.

## Supporting information

**S1 File. PRISMA-P checklist.**
(DOC)

**S2 File. Instructions for search flow.**
(DOCX)

**S3 File. Novel tool.**
(XLSX)

## Author Contributions

**Conceptualization:** Dustin W. Davis, James W. Navalta.

**Methodology:** Dustin W. Davis, Bryson Carrier, Brenna Barrios, Kyle Cruz, James W. Navalta.

**Project administration:** Dustin W. Davis, James W. Navalta.

**Resources:** Dustin W. Davis.

**Software:** Dustin W. Davis.

**Supervision:** Dustin W. Davis, James W. Navalta.

**Validation:** Dustin W. Davis, Bryson Carrier, Brenna Barrios, Kyle Cruz, James W. Navalta.

**Visualization:** Dustin W. Davis.

**Writing – original draft:** Dustin W. Davis, Bryson Carrier, Brenna Barrios, Kyle Cruz, James W. Navalta.

**Writing – review & editing:** Dustin W. Davis, Bryson Carrier, Brenna Barrios, Kyle Cruz, James W. Navalta.

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
