## [Decision Letter · Decision Letter 0]

20 Aug 2021

PONE-D-21-12017

A protocol and novel tool for systematically reviewing the effects of mindful walking on mental and cardiovascular health

PLOS ONE

Dear Dr.  Dustin,

Thank you for submitting your manuscript to PLOS ONE. After careful consideration, we feel that it has merit but does not fully meet PLOS ONE’s publication criteria as it currently stands. Therefore, we invite you to submit a revised version of the manuscript that addresses the points raised during the review process.

Your manuscript requires MINOR Revisions based on the comments from two reviewers. Please ensure that in your revised manuscript you address all comments.

I looked forward to the revised manuscript

submit your revised manuscript by 12th September 2021. If you will need more time than this to complete your revisions, please reply to this message or contact the journal office at plosone@plos.org. Please include the following items when submitting your revised manuscript:

We look forward to receiving your revised manuscript.

Kind regards,

Prof. Giuseppe Filiberto Serraino, M.D., Ph.D.

Academic Editor

PLOS ONE

Journal Requirements:

Reviewers' comments:

Reviewer's Responses to Questions

**Comments to the Author**

1. Does the manuscript provide a valid rationale for the proposed study, with clearly identified and justified research questions?

Reviewer #1: Yes

Reviewer #2: Yes

2. Is the protocol technically sound and planned in a manner that will lead to a meaningful outcome and allow testing the stated hypotheses?

Reviewer #1: Yes

Reviewer #2: Yes

3. Is the methodology feasible and described in sufficient detail to allow the work to be replicable?

Reviewer #1: Yes

Reviewer #2: Yes

4. Have the authors described where all data underlying the findings will be made available when the study is complete?

Reviewer #1: Yes

Reviewer #2: Yes

5. Is the manuscript presented in an intelligible fashion and written in standard English?

Reviewer #1: Yes

Reviewer #2: Yes

6. Review Comments to the Author

You may also provide optional suggestions and comments to authors that they might find helpful in planning their study.

Reviewer #1: The review protocol is good and will bring perfect conclusions where the communities will be benefitec.

Reviewer #2: Review of PONE-D-21-12017: A protocol and novel tool for systematically reviewing the effects of mindful walking on mental and cardiovascular health

This article presents a step-by-step protocol for conducting a systematic review of studies examining the effects of mindful walking on mental and cardiovascular health to ensure procedural transparency and replication of the protocol in future reviews on the topic. The authors also present a novel, free-of-charge tool they have developing for tracking the sources that are identified and screened for inclusion.

The authors present a convincing case for the need for a systematic review in this field, and for the value of providing a protocol to ensure procedural transparency and facilitate replication of the protocol in future reviews. Together with the novel source-tracking tool they offer, their study makes an important contribution to the literature in the specific field, and to the conducting of systematic reviews more generally.

I have a few minor comments and clarifications to request of the authors.

Clarifications:

1. Lines 109-113: Most meta-analyses include studies with interventions and populations that differ. The heterogeneity is something that can be evaluated statistically to determine whether or not it is appropriate to use meta-analysis (with random or fixed effects); and/or stratified meta-analyses can be conducted including more similar intervention types or population groups. I don’t know if the appropriateness of meta-analysis can/should be determined in advance at the stage of the protocol, presumably before the literature search has been conducted (so before knowing the number and characteristics of studies to be included). If, however, the inappropriateness of conducting a meta-analysis could be determined by the preliminary scoping searches mentioned in lines 95-96, it would seem to me that it would be better to state that as the reason for deciding at the protocol stage that meta-analysis will not be appropriate.

2. In Table 1, I don’t understand why the items “Frequency, intensity…”, “Location of the intervention”, and “Participant’ age, sex…” are included under Outcomes. Perhaps this information should be included only in the text, rather than in the PICOS table, or another category should be added to the PICOS table (e.g., Intervention and Participant characteristics).

Minor comments:

1. Lines 92-93: something is missing in this sentence (should perhaps be “we explain the novel tool that was created…”).

2. In the novel tool spreadsheet, the link did not work.

3. The reference management software, RefWorks, that the authors used, is not a free resource; so they may want to suggest alternative free/open-source reference management softwares to complement their free tool.

7. PLOS authors have the option to publish the peer review history of their article (what does this mean?). If published, this will include your full peer review and any attached files.

Reviewer #1: **Yes: **THOMAS AYALEW ABEBE

Reviewer #2: No

---

## [Author Response · Author response to Decision Letter 0]

26 Aug 2021

Thank you for your valuable feedback. We uploaded a file named "Response to Reviewers" that contains our letter and point-by-point responses to the editor and reviewers. We also pasted our point-by-point responses below.

Reviewer #2’s Clarifications and Comments and Our Responses

Clarifications

1. “Lines 109-113: Most meta-analyses include studies with interventions and populations that differ. The heterogeneity is something that can be evaluated statistically to determine whether or not it is appropriate to use meta-analysis (with random or fixed effects); and/or stratified meta-analyses can be conducted including more similar intervention types or population groups. I don’t know if the appropriateness of meta-analysis can/should be determined in advance at the stage of the protocol, presumably before the literature search has been conducted (so before knowing the number and characteristics of studies to be included). If, however, the inappropriateness of conducting a meta-analysis could be determined by the preliminary scoping searches mentioned in lines 95-96, it would seem to me that it would be better to state that as the reason for deciding at the protocol stage that meta-analysis will not be appropriate.”

Our Response: Thank you for this important feedback. We changed the phrasing of lines 109-114. We no longer state we will not conduct a meta-analysis. Instead, we state we did not plan a meta-analysis because the scoping searches revealed clinical heterogeneity among the relevant studies (i.e., population, intervention, comparison groups, outcomes, study settings).

2. “In Table 1, I don’t understand why the items “Frequency, intensity…”, “Location of the intervention”, and “Participant’ age, sex…” are included under Outcomes. Perhaps this information should be included only in the text, rather than in the PICOS table, or another category should be added to the PICOS table (e.g., Intervention and Participant characteristics).”

Our Response: We agree that the text you mentioned would be more appropriate in a different place in Table 1. We moved the participant characteristics we will extract to the population row and the intervention characteristics to the intervention row.

Minor Comments

1. “Lines 92-93: something is missing in this sentence (should perhaps be “we explain the novel tool that was created…”).”

Our Response: We corrected the sentence. Thank you.

2. “In the novel tool spreadsheet, the link did not work.”

Our Response: We apologize for providing you with a link that did not work. The new link that works is here. The link takes you to the same file you already viewed, S3_File, that is housed on Google Drive. The other supplemental files are available in the same Google Drive folder that houses the S3_File.

3. “The reference management software, RefWorks, that the authors used, is not a free resource; so they may want to suggest alternative free/open-source reference management softwares to complement their free tool.”

Our Response: This is a good point. We updated the S2_File (the only place where RefWorks is mentioned) to suggest Mendeley and Zotero as alternative free options.

---

## [Editor Report · Decision Letter 1]

28 Sep 2021

A protocol and novel tool for systematically reviewing the effects of mindful walking on mental and cardiovascular health

PONE-D-21-12017R1

Dear Dr. Davis,

We’re pleased to inform you that your manuscript has been judged scientifically suitable for publication and will be formally accepted for publication once it meets all outstanding technical requirements.

Kind regards,

Giuseppe Filiberto Serraino, M.D., Ph.D.

Academic Editor

PLOS ONE

---

## [Editor Report · Acceptance letter]

1 Oct 2021

PONE-D-21-12017R1 

A protocol and novel tool for systematically reviewing the effects of mindful walking on mental and cardiovascular health 

Dear Dr. Davis:

I'm pleased to inform you that your manuscript has been deemed suitable for publication in PLOS ONE. Congratulations! Your manuscript is now with our production department. 

Kind regards, 

on behalf of

Professor Giuseppe Filiberto Serraino 

Academic Editor

PLOS ONE